

# Fine-scale differentiation between *Bacillus anthracis* and *Bacillus cereus* group signatures in metagenome shotgun data

Robert A. Petit III[1], James M. Hogan[2], Matthew N. Ezewudo[1], Sandeep J. Joseph[1] and Timothy D. Read[1]

[1] Department of Medicine, Division of Infectious Diseases, Emory University School of Medicine, Atlanta, GA, United States of America
[2] Queensland University of Technology, Brisbane, Australia

Corresponding author
Timothy D. Read, tread@emory.edu

**OPEN ACCESS**

## ABSTRACT

**Background**. It is possible to detect bacterial species in shotgun metagenome datasets through the presence of only a few sequence reads. However, false positive results can arise, as was the case in the initial findings of a recent New York City subway metagenome project. False positives are especially likely when two closely related are present in the same sample. *Bacillus anthracis*, the etiologic agent of anthrax, is a high-consequence pathogen that shares >99% average nucleotide identity with *Bacillus cereus* group (BCerG) genomes. Our goal was to create an analysis tool that used k-mers to detect *B. anthracis,* incorporating information about the coverage of BCerG in the metagenome sample.

**Methods**. Using public complete genome sequence datasets, we identified a set of 31-mer signatures that differentiated *B. anthracis* from other members of the *B. cereus* group (BCerG), and another set which differentiated BCerG genomes (including *B. anthracis*) from other *Bacillus* strains. We also created a set of 31-mers for detecting the lethal factor gene, the key genetic diagnostic of the presence of anthrax-causing bacteria. We created synthetic sequence datasets based on existing genomes to test the accuracy of a k-mer based detection model.

**Results**. We found 239,503 *B. anthracis*-specific 31-mers (the *Ba31 set*), 10,183 BCerG 31-mers (the *BCerG31 set*), and 2,617 lethal factor k-mers (the *lef31* set). We showed that false positive *B. anthracis* k-mers—which arise from random sequencing errors—are observable at high genome coverages of *B. cereus*. We also showed that there is a "gray zone" below 0.184× coverage of the *B. anthracis* genome sequence, in which we cannot expect with high probability to identify lethal factor k-mers. We created a linear regression model to differentiate the presence of *B. anthracis*-like chromosomes from sequencing errors given the BCerG background coverage. We showed that while shotgun datasets from the New York City subway metagenome project had no matches to *lef31* k-mers and hence were negative for *B. anthracis*, some samples showed evidence of strains very closely related to the pathogen.

**Discussion**. This work shows how extensive libraries of complete genomes can be used to create organism-specific signatures to help interpret metagenomes. We contrast "specialist" approaches to metagenome analysis such as this work to "generalist" software that seeks to classify all organisms present in the sample and note the more general utility of a k-mer filter approach when taxonomic boundaries lack clarity or high levels of precision are required.

# INTRODUCTION

There is great interest in the use of shotgun metagenome data to detect pathogens in clinical and environmental samples. A large number of bioinformatic tools have been developed (*McIntyre et al., 2017*) that use different algorithmic approaches to rapidly parse and analyze sequence data files. Over the last 8–10 years, these data have been generated primarily by Illumina sequencing technology. Typically, sequences from metagenomic data files are matched against public reference databases, such as NCBI RefSeq. Consistency of matches across the tree of life is dependent therefore on the database entries being correctly labelled, having similar levels of representation across species, and having species defined in a consistent manner. However, we are beginning to understand how the skewed representation of taxa contained in the database sometimes affects sampling accuracy (*Nasko et al., 2018*). The classification of many bacterial species harks back to distinctions based on morphological, biochemical and virulence characteristics, made prior to the advent of DNA sequencing. Sometimes, unusually close species boundaries can confound metagenomic classifiers and result in false positive matches. In 2015, *Afshinnekoo et al. (2015a)* published initial findings from an extensive study of the New York Subway metagenome, which claimed that they had detected bacteria responsible for anthrax (*Bacillus anthracis*) and plague (*Yersinia pestis*). While these misidentifications were swiftly corrected (*Mason, 2015*; *Afshinnekoo et al., 2015b*), indistinct or fuzzy boundaries between species may yield many errors of this nature.

   *B. anthracis*, the pathogen that is the focus of this work, is a Gram-positive bacterium that forms tough endospores allowing it to survive dormant in the environment for years. The 5.2 (Mbp) main chromosome shares an average nucleotide identity (ANI, *Konstantinidis & Tiedje, 2005*) in excess of 99% with other members of the collection of species known as the '*Bacillus cereus* group' (*BCerG*) (*Helgason et al., 2000*). The most common species in this group are *B. cereus*, *B. thuringiensis* and *B. mycoides* (*Helgason et al., 2000*; *Zwick et al., 2012*). The recommended level of difference between bacterial species is an ANI of 95% (*Konstantinidis & Tiedje, 2005*). While BCerG strains are mostly opportunistic pathogens of invertebrates and are commonly found in soil, *B. anthracis* kills mammals (*Carlson et al., 2018*). Spores are generally found at high titers in soils where animals have recently died from anthrax. Phylogeographic analysis has shown that *B. anthracis* is probably native to Africa, with only recent transfer of a limited number of lineages to other continents (*Keim & Wagner, 2009*). For these reasons, it would be an unusual outcome to find spores in the New York subway (*Ackelsberg et al., 2015*)

   What sets *B. anthracis* apart from other BCerG strains is the presence of two plasmids: pXO1 (181 kb), which carries the lethal toxin genes, and pXO2 (94 kb), which includes genes for a protective capsule. Without either of these plasmids, *B. anthracis* is considered attenuated in virulence and unable to cause classic anthrax (*Dixon et al., 1999*). Plasmids from other BCerG genomes may be very similar to pXO1 and pXO2 but lack the important

virulence genes (*Rasko et al., 2007*). Rarely, BCerG strains carry pXO1 and appear to cause anthrax-like disease (*Hoffmaster et al., 2004*; *Hoffmann et al., 2017*); pXO2-like plasmids are also quite common in BCerG and other *Bacillus* species (*Pannucci et al., 2002*; *Cachat et al., 2008*).

Shortly after the release of the NYC subway metagenome paper, we produced a blog post (*Petit III et al., 2015*) that critically re-analyzed these data in the light of what was known about *B. anthracis* genomics. This work, and other critiques, led to reassessment of the data and revisions to the original manuscript. In this paper, we incorporate some of the results introduced informally on our blog and extend them to create a k-mer based approach—using recent public *B. anthracis* and BCerG data—to analyze in greater detail how to search for traces of *B. anthracis* in shotgun metagenome data. While elements of this method are necessarily specific to *B. anthracis* and the context of the BCerG group, the general strategy has far broader utility and this work is a model for future "specialist" studies based on k-mer filtering.

## METHODS

### Metagenome data and reference genome sequences

Shotgun metagenomic data from the "NYC" study SRP051511 (*Afshinnekoo et al., 2015a*) were downloaded from the Sequence Read Archive (SRA) with sra-tools (fastq-dump -I $SRA_ACCESSION, v2.8.2, https://github.com/ncbi/sra-tools). Reference genomes for different taxonomic groups were downloaded from the NCBI Nucleotide database in April 2018 with the following queries:

All BCerG genomes = 'txid86661[Organism:exp] AND "complete genome"[Title] AND refseq[filter] AND 3000000:7000000[Sequence Length]'

All non-BCerG *Bacillus* genomes = 'txid1386[Organism:exp] NOT txid86661 [Organism:exp] "complete genome"[Title] AND 3000000:7000000[Sequence Length] AND refseq[filter]'

*B. anthracis* genomes were included in the BCerG genome query. The lethal factor gene was extracted from completed pXO1 plasmids downloaded with the following query:

pXO1 plasmid = 'pXO1[Title] AND 140000:200000[Sequence Length] '

The results of these queries, as of April 2018, are available on our git repository.

### Mapping metagenome data to *B. anthracis* plasmids and chromosomes

*B. anthracis* positive samples and control samples were mapped against reference pXO1 (CP009540) and pXO2 (NC_007323) plasmids and reference *B. anthracis* (CP009541) and *B. cereus* (NC_003909) completed genomes with BWA (bwa mem -t $NUM_CPU $REFERENCE $FASTQ_R1 $FASTQ_R2 >$SAM_FILE, v0.7.5a-r405, *Li & Durbin, 2009*). The aligned reads in SAM format were converted to sorted BAM and indexed with SAMtools (samtools view -@ 10 -bS $SAM_FILE |samtools sort -@ 10 - $SAMPLE, v1.1,

*Li et al., 2009*). The per base coverage was extracted with genomeCoverageBed from BEDTools (genomeCoverageBed -d -ibam $BAM_FILE |gzip –best - >$COVERAGE, v2.16.2, *Quinlan & Hall, 2010*). Coverage across the plasmids and chromosomes was plotted for multiple sliding windows with a custom Rscript. Mapped reads were extracted and saved in FASTQ with bam2fastq (bam2fastq -o $FASTQ –no-unaligned $BAM_ FILE, v1.1.0, https://gsl.hudsonalpha.org/information/software/bam2fastq) and FASTA format with fastq_to_fasta from FASTX Toolkit (cat $FASTQ_FILE |fastq_to_fasta -Q33 -n |gzip –best - >$FASTA_OUTPUT, v0.0.13.2, *Gordon & Hannon, 2010*). Scripts, runtime parameters, and output are available at this site (*Petit III et al., 2015*): https://github.com/Read-Lab-Confederation/nyc-subway-anthrax-study.

### Custom 31-mer assay for *B. anthracis* and *Bacillus cereus* Group

In preliminary analysis we found four BCerG genomes misclassified in the NCBI Taxonomy database as not being part of the BCerG (see the Results section). To create a rational method to assign taxonomy to genomes for this study we used mash (mash sketch -k 31 -s 100000 -p $NUM_ CPU -o $OUTPUT_PREFIX *.fasta, v2.0, *Ondov et al., 2016*) to reclassify mislabeled *Bacillus* genomes as *B. anthracis*, non-*anthracis* BCerG, or non-BCerG. We identified *B. anthracis* strain 2002013094 (NZ_CP009902) as the most distant (Mash distance 0.000687) *B. anthracis* member from *B. anthracis* str. Ames (NC_ 003997). We also identified *B. cytotoxicus* NVH 391-98 (NC_009674) as the most distant (Mash distance 0.135333) BCerG member from *B. anthracis* str. Ames (NC_003997). We then determined the Mash distance of all *Bacillus* genomes from *B. anthracis* str. Ames (mash dist -p $NUM_CPU $MASH_SKETCH $FASTA_FILE). We used the Mash distance to reclassify each Bacillus genome as *B. anthracis* (Mash distance ≤ 0.000687), non-*anthracis* BCerG (Mash distance ≤ 0.135333), or non-BCerG (Mash distance > 0.135333). A phylogeny of all completed Bacillus genomes was created with mashtree (mashtree –numcpus 20 *.fasta >bacillus-mashtree.dnd, v0.32, https://github.com/lskatz/mashtree).

Sequence 31-mers were extracted and counted with Jellyfish (jellyfish count -C -m 31 -s 1M -o $JELLYFISH_DB $FASTA_ FILE, v2.2.3, *Marçais & Kingsford, 2011*) and partitioned into two distinct sets characteristic of BCerG (BCerG31) and *B. anthracis* (Ba31) (Fig. 1). The BCerG31 and Ba31 sets were initially comprised of 31-mers conserved within *every* member of BCerG (including *B. anthracis*) and those restricted to only *B. anthracis*, respectively. Any Ba31-mers found in non-*anthracis* BCerG members or non-BCerG genomes were filtered out. Likewise, any BCerG31-mers found in non-BCerG Bacillus genomes were filtered out. 31-mers found in rRNA were filtered out with a Jellyfish database created from the SILVA rRNA database (*Quast et al., 2013*). We further filtered the Ba31 and BCerG31 sets using the non-redundant nucleotide sequence database (NT v5, downloaded April 2017). We used BLASTN (blastn -max_hsps 1 -max_target_ seqs 1 -dust no -word_size 7 -outfmt 15 -query $FASTA_FILE -db $BLAST_DB -evalue 10000 -num_threads $NUM_CPU, v2.8.0, *Camacho et al., 2009*) to align Ba31 against non-*anthracis* BCerG sequences and BCerG31 against non-BCerG sequences. 31-mers with an exact match were filtered out.

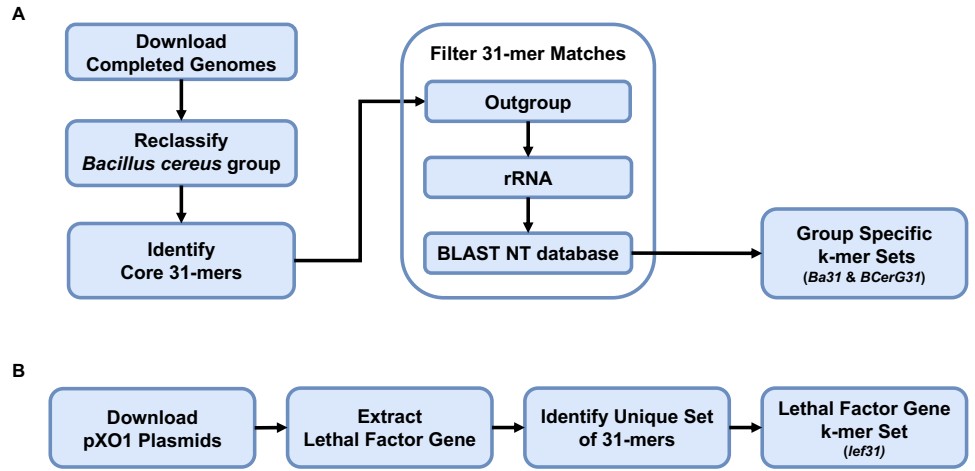

**Figure 1** **Flowchart of strategy for primer design.** We developed a strategy for selecting the Ba31 and BCerG31 (A) and lef31 (B) k-mer sets. In (A) the outgroup is determined by the k-mer set. For Ba31, the outgroup was comprised of all the non-*B. anthracis* genomes; for BCerG31, it consisted of all non-*B. cereus* group genomes.

## Finding the limits for lethal factor-based detection of *B. anthracis*

We used *B. anthracis* whole genome shotgun sequencing projects to determine the limit of detection of lethal factor k-mers (lef31). We defined lef31 as the unique set of 31-mers identified in *lef* genes downloaded from the NCBI Nucleotide database (previously described) (Fig. 1). *B. anthracis* projects were identified from the SRA with the following query:

*B. anthracis* projects = 'genomic[Source] AND random[Selection] AND txid86661 [Organism:exp] AND paired[Layout]) AND wgs[Strategy] AND "Illumina HiSeq"'

In this work we have assumed a 95% 'confidence limit' for detection of the lethal factor k-mers, so that detection is held to fail if fewer than 95% of a set of random subsamples are found to contain *at least one* lethal factor k-mer. The threshold is then obtained through computational experiment. For each project, we started at 0.2× *B. anthracis* genome coverage and extracted 100 random subsamples of sequences, using Jellyfish as before to determine if at least one lethal factor k-mer was present. We then continued this process, reducing the coverage until fewer than 95% of the subsamples contained at least one lethal factor k-mer. The previous coverage was then recorded as the limit of detection of the lethal toxin for a given sample.

## Assessing Quality of *B. anthracis* and *B. cereus* Group specific 31-mers

We used ART (art_illumina -l 100 -f $COVERAGE -na -ss HS20 -rs $RANDOM_SEED -i $FASTA_FILE -o $OUTPUT_PREFIX, vMountRainier-2016-06-05, *Huang et al., 2012*) to simulate 100 bp reads with the built-in Illumina HiSeq 2000 error model for each non-*anthracis Bacillus* genome. The Illumina HiSeq 2000 error model was selected to match

the predominant sequencing technology of the NYC dataset. We simulated coverages ranging from 0.01× to 15× to determine if false positive Ba31 matches were uniform across non-*anthracis* BCerG members. We counted 31-mers for each simulated read set with Jellyfish as previously described. We then used the k-mer counts to determine the false positive Ba31 counts in non-anthracis genomes. We found the false positive Ba31 counts to be higher in non-*B. anthracis* genomes that were most closely related to *B. anthracis* (please see results section). A subset of non-*B. anthracis* BCerG genomes with a Mash distance less than 0.01 from *B. anthracis*, previously described, were selected as our model set. We further simulated coverages from 15× to 100× to match levels of coverage observed in the NYC dataset. We then applied linear regression, implemented in the R base stats package (R v3.4.3), on this subset to develop a predictive model with the observed Ba31 count as our dependent variable and the observed BCerG k-mer coverage as our independent variable.

### Prediction of low coverage *B. anthracis* chromosome in simulated metagenomic sequencing datasets

We used ART to simulate metagenomic mixtures of *B. anthracis* str. Ames (NC_003997) and *B. cereus* strain JEM-2 (NZ_CP018935). *B. cereus* strain JEM-2 was selected because it was the closest non-*anthracis* BCerG member to *B. anthracis* str. Ames (Mash distance 0.00873073). We used coverages between 0–100× for *B. cereus* and coverages between 0–0.2× for *B. anthracis*. A python script (subsample-ba-lod.py NC_003997.fasta NZ_CP018935.fasta coverages-ba.txt coverages-bcg.txt temp_folder/ fasta/ ba-specific-kmers.fasta bcg-specific-kmers.fasta) was created to simulate mixtures for each pairwise combination of *B. cereus* and *B. anthracis* coverages. For each mixture, we determined the Ba31 and BCerG31 counts with Jellyfish as previously described. This process was repeated 20 times per pairwise combination of coverages. The model was applied to determine what level of *B. anthracis* coverage was required to differentiate observed Ba31-mers from sequencing errors.

We determined Ba31, BCerG31 and lef31 counts for each sample in the NYC study. The model was applied to these counts to determine if observed *B. anthracis* k-mers exceeded the level expected due to sequencing errors.

We processed each of the subsampled mixtures and samples from the NYC study with KrakenHLL (krakenhll –report-file $REPORT_FILE –db $DATABASE >$SEQUENCES, v0.4.7, *Breitwieser & Salzberg, 2018*). We used dustmasker (dustmasker -outfmt fasta, v2.8.0, *Camacho et al., 2009*) to create a DUST-masked version of the standard Kraken database (kraken-build –standard –db $DATABASE, database built in April 2017) for this analysis. From the final Kraken report, the number of reads and unique k-mers identified for *B. anthracis* were extracted. We compared these results to our method.

Output, figures, runtime parameters and scripts from this study are available in a Git repository hosted at: https://doi.org/10.5281/zenodo.1323741.

## RESULTS

### NY subway metagenome sequences map to core regions of *B. anthracis* and *B. cereus* chromosome and plasmids but not to lethal factor gene

In the original analysis of the subway metagenome (*Afshinnekoo et al., 2015a*), two samples (P00134 (SRR1748707, SRR1748708), and P00497 (SRR1749083)) were reported to contain reads that mapped to *Bacillus anthracis* based on results obtained using the Metaphlan software (*Segata et al., 2012*). We found that 792,282 reads from P00134 and 270,964 reads from P00497 mapped to the *B. anthracis* strain Sterne chromosome. The reads aligned along the entire length of the chromosome, forming a characteristic peak at the replication origin, a pattern often seen when other bacterial chromosomes have been recovered from metagenome samples (*Brown et al., 2016*). However, a similar number of reads from P00134 and P00497 (765,466 reads and 265,776 reads, respectively) mapped to the *B. cereus* 10987 chromosome. We also found that P00134 and P00497 reads mapped to the both the pXO1 and pXO2 plasmids in conserved "backbone" regions (*Rasko et al., 2007*) but that no read mapped to the mobile element containing the *lef* lethal factor gene. These results showed that the close taxonomic relationship of *B. anthracis* and BCerG made identification of the biothreat agent by mapping reads alone unreliable. In addition, the pXO1 and pXO2 plasmids were not reliable as positive markers for *B. anthracis* at low genome coverages (when the *lef* gene may not be sampled, see next section) because backbone sequences cross-matched against plasmids found in BCerG strains.

### *B. anthracis* genome coverage below 0.184× is a "gray area" for detection, where lethal toxin genes may not be sampled

The best test for presence of virulent *B. anthracis* (or virulent *B. cereus* strains containing pXO1) is detection of the lethal factor gene (2,346 bp) (*Bragg & Robertson, 1989*). However, at low sequence coverage of the pathogen, it is not certain that reads from this gene will be present (given the 3:1 copy number ratio of pXO1 to *B. anthracis* chromosome (*Read et al., 2002*) the ratio of chromosome to *lef* is ∼620:1). We identified 2,617 31-mers present in 36 *lef* gene sequences and called this set "lef31". To estimate the coverage sufficient that we would expect (with probability above some threshold value, here 0.95) to observe lethal factor sequences, we randomly subsampled reads from 164 *B. anthracis* genome projects and tested for the presence of *at least one* lef31 match (Fig. 2). With Ba31 and BCerG31 k-mer coverages below 0.103× and 0.112×, respectively, this analysis showed we would have less than a 95% chance of sampling a single lef31 k-mer *even if the lef gene were present*. These k-mer coverages are approximately 0.184×-fold *B. anthracis* genome coverage, or 9,360 100 base pair reads (Fig. S1).

### Conserved and specific 31-mer sets for *B. anthracis* and BCerG chromosomes

The results of the previous section showed that at low *B. anthracis* genome coverage, detection of the lethal factor is not guaranteed. In metagenomic samples, in which sequencing coverage is expected to be low for rare organisms, the most reliable way

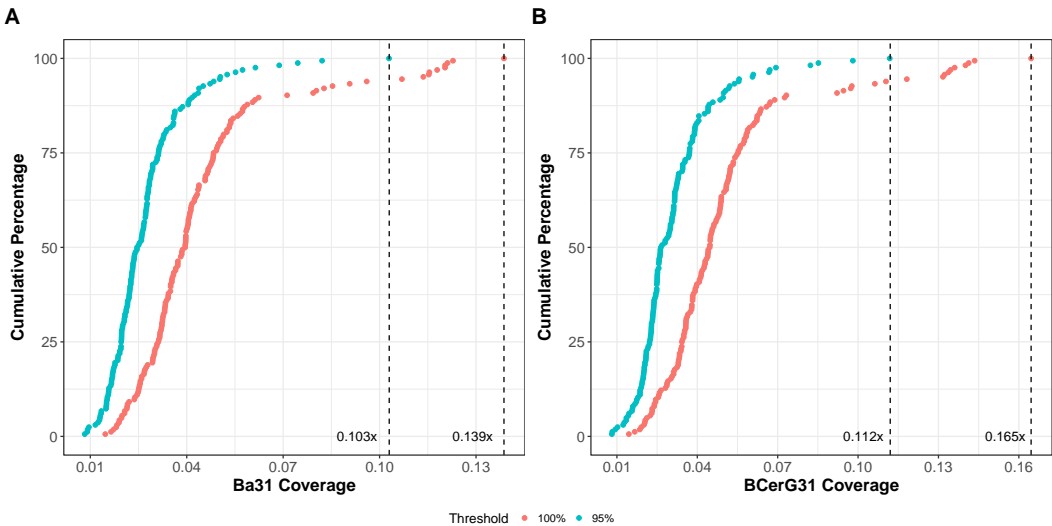

**Figure 2** **Limit of detection for lethal factor gene k-mers (lef31).** A total of 164 *B. anthracis* sequencing projects were subsampled to different levels of genome coverage, with 100 random subsamples obtained for each coverage level. Our ability to detect the lethal factor gene is assessed by considering the number of these subsamples for which we find at least one lef31 k-mer hit. Two thresholds—95% and 100%—were employed and are shown as colored series below.The figure thus shows the percentage of the *B. anthracis* sequencing projects for which 95% (or 100%) of the random subsamples contain at least one lef31 k-mer. (A) shows results with respect to Ba31 k-mer coverage while (B) shows the corresponding results for BCerG coverage. The vertical dashed lines show the coverage limits for detection at the respective threshold levels.

to detect *B. anthracis* was to use chromosomal genetic signatures that distinguished the species from close relatives. We identified 239,503 31-mers conserved in 48 *B. anthracis* reference genomes that were not also detected in the remainder of the *Bacillus* genus (331 genomes), rRNA sequences, or the BLAST non-redundant nucleotide database. We called this set "Ba31".

We created a second set of 31-mers specific to and conserved in all BCerG genomes (including *B. anthracis*). Surprisingly, our initial analysis produced zero 31-mers specific to all 139 BCerG strains and not other *Bacillus*. Inspection of the whole genome phylogeny (Fig. 3) showed that four genomes (NZ_CP007512, NZ_CP017016, NZ_CP020437, NZ_CP02512) that fell within the BCerG clade based on phylogeny had not been classified as BCerG in the NCBI Taxonomy hierarchy. After reclassifying these strains as BCerG, we identified 10,183 BCerG specific 31-mers, which we called "BCerG31".

## High background levels of *B. cereus* strains produce false positive *B. anthracis* specific k-mers due to random sequence errors

We simulated synthetic data of *Bacillus* reference genomes at different genome coverages using ART software with an error model based on Illumina short read data (*Huang et al., 2012*) (Fig. 4). We defined 'k-mer coverage' as the sum of counts for k-mers detected divided by the number of k-mers in the k-mer set. Ba31 and BCerG k-mer coverage had a linear relationship with genome coverage (Fig. S1). The coefficient was less than 1

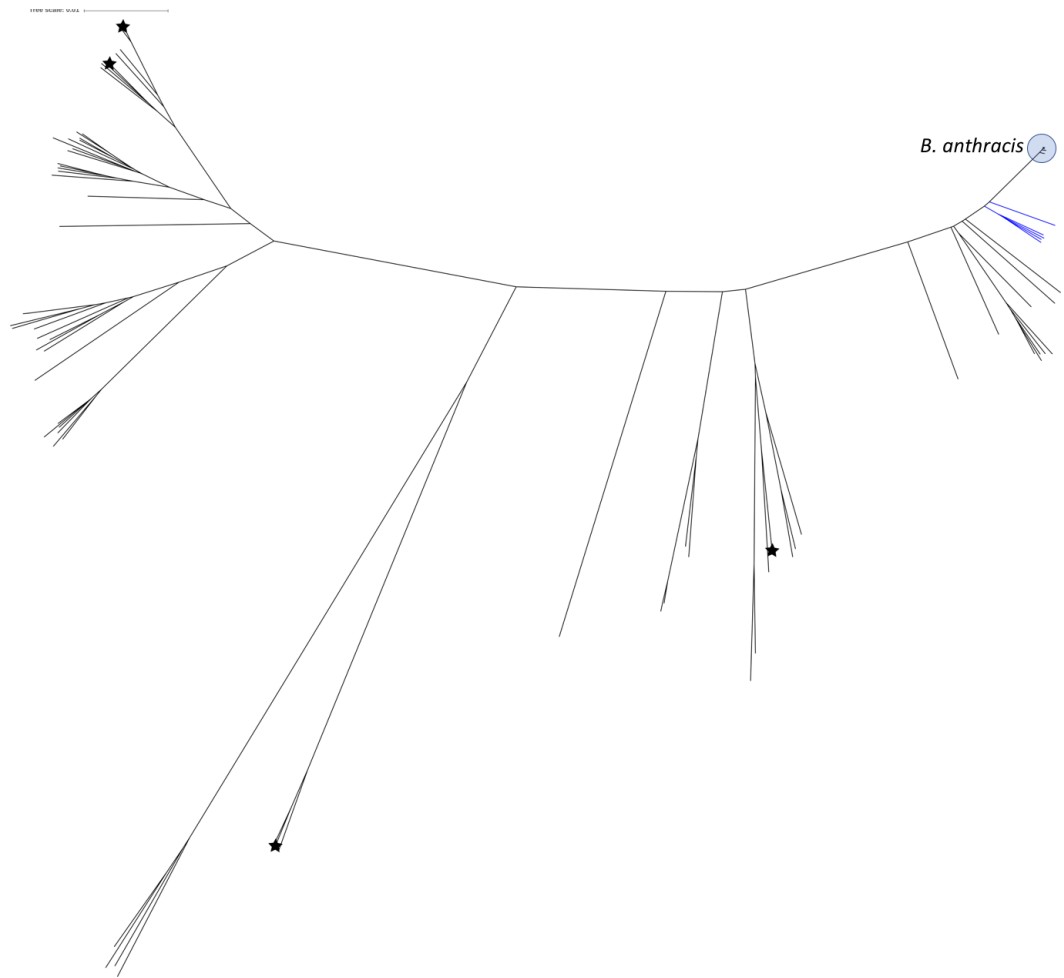

**Figure 3** **Unrooted phylogeny of BCerG genome assemblies used in the study after reclassifying BCerG strains.** An unrooted phylogenetic representation of 140 BCerG genomes using Mashtree (v0.32, https://github.com/lskatz/mashtree). Genomes reclassified as BCerG members with mash (v2.0, *Ondov et al., 2016*) are indicated with stars. The clade colored blue are *B. cereus* genomes closely related to *B. anthracis* that were used to model false positive results (Fig. 4).

(0.56 and 0.61 for Ba31 and BCerG31 respectively), because some portions of the chromosomes were not well sampled by the k-mers. We found a strong linear relationship between Ba31 coverage and BCerG31 coverage within *B. anthracis* genome subsamples (Pearson's Correlation $r = 0.99$, $p < 0.001$, Fig. S2). As expected, the same relationship did not appear when we subsampled non-*B. anthracis* BCerG members (Pearson's Correlation $r = 0.74$, $p < 0.001$, Fig. S3). However, we did see a small number of Ba31 k-mers detected, which we suspected were due to random errors introduced by Illumina sequencing (Fig. 4). The counts of false positive Ba31 k-mers scaled with the approximate genetic distance to *B. anthracis* (as measured by mash *Ondov et al., 2016*) (Fig. S4). We simulated synthetic data for a group of BCerG strains most closely related to *B. anthracis* (Fig. 3). We developed a linear regression model to relate BCerG k-mer coverage and sequencing errors based on
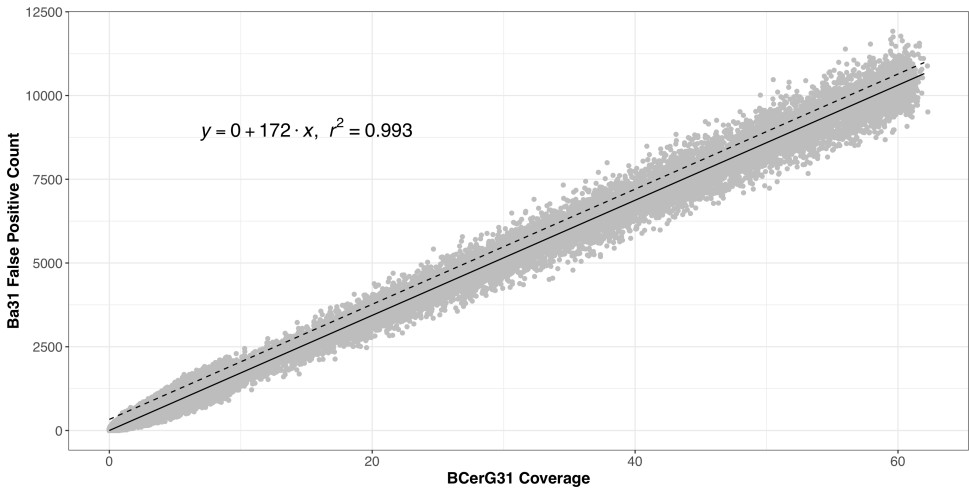

$y = 0 + 172 \cdot x, \quad r^2 = 0.993$

**Figure 4  Linear regression model fit of BCerG coverage and false positive Ba31 counts.** We created random synthetic FASTQ files based on BCerG chromosomes from the clade closest to *B. anthracis* (blue in Fig. 3) at different genome coverages and counted the false positive Ba31 k-mers. Shown is the fit of a linear regression model with an intercept of 0, with BCerG31 coverage as the independent variable and the Ba31 false positive count as the dependent variable. The solid line shows the predicted values from the model, and the dashed line reflects the upper 99% prediction interval for the parameters, which we use in the analyses above.

this group (Fig. 4). For every unit of BCerG31 k-mer coverage, we predicted 172 Ba31 false positive k-mer counts.

## A "specialist" model to interpret patterns of *B. anthracis* genetic signatures in metagenome samples

In real metagenome samples *B. anthracis*, if present, may only account for a low proportion of the total reads and may also be mixed with higher proportions of closely related BCerG strains. We sought to use the k-mer sets developed in the previous sections and knowledge of the *lef* gray zone coverage and BCerG false positive rate to interpret both synthetic and real metagenome datasets. The logic for assignment is shown in Table 1 and Fig. S5.

For our synthetic dataset we mixed low coverage *B. anthracis* with higher coverages of BCerG sequence data (see Methods). We calculated the BCerG31 and Ba31 coverages for each mixture. Based on the BCerG sequence error model, we calculated the 99% count of Ba31 signatures predicted to be present by sequencing error under the assumption that there was no *B. anthracis* present and that all BCerG were drawn from the most closely related clade (Fig. 3). We also reported whether the Ba31 coverage lay in or above the gray zone (Table 2, File S2, Fig. S6). When *B. anthracis* was below 0.003× genome coverage (approximately 16,000 bp), we could not distinguish its presence from errors produced in the absence of *B. cereus*. As expected, we found that the level of BCerG coverage determined the lower limit to differentiate genuine Ba31 hits from sequencing errors. At 75× BCerG coverage the required *B. anthracis* coverage to differentiate Ba31 matches from sequencing errors doubled to 0.006×. The threshold for accurate detection was further raised to 0.01× *B. anthracis* genome coverage at 100× BCerG coverage.

**Table 1 Potential outcomes of *B. anthracis* detection, given matches to the Ba31 set in a shotgun metagenome dataset.** This table discusses the interpretation of four cases when Ba31 k-mer matches are found in the dataset. Columns 1–3 are; lef31 match; whether Ba31 coverage is in the Gray Zone; and whether Ba31 coverage is above the 99% of the error model based on BCerG coverage.

| Case | Lef31 | Gray Zone | Exceeds 99% P.I.[a] | Interpretation |
|------|-------|-----------|---------------------|----------------|
| 1 | yes | yes or no | yes or no | Evidence of lethal factor gene, could be *B. anthracis* or a *B. cereus* strain carrying the pXO1 plasmid. |
| 2 | no | yes | yes | Possible *B. anthracis* or closely related strain based on high Ba31 counts but genome coverage too low to guarantee seeing the *lef* gene. Requires more sequence coverage and/or validation by PCR or other methods. |
| 3 | no | no | yes | Ba31 matches exceed what is expected by the BCerG error model, but are at a level of genome coverage at which lethal factor should have been detected. Most likely explanation is *B. anthracis* strain cured of pXO1 or unsequenced lineage closely related to *B. anthracis*. |
| 4 | no | yes or no | no | Most likely scenario is that BCerG background produced Ba31 k-mers through random errors but impossible to also rule out presence of low coverage *B. anthracis* |

**Notes.**
[a]Prediction interval.

In contrast, when the samples were classified using KrakenHLL (*Breitwieser & Salzberg, 2018*), an accurate generalist program based on 31-mers, we found that all were predicted to contain *B. anthracis*, including negative controls (Table 2). The *B. anthracis* calls were made because of the sequence errors from the high coverage BCerG genomes.

Finally, we tested our model against the NYC dataset (Table 3, File S2). All 1,458 samples were negative for lef31, in line with the conclusion reached from re-analysis of the dataset that *B. anthracis* was absent from the NY subway (*Mason, 2015*). We found that 1,367 of the 1,458 samples had at least one BCerG31 k-mer match and, of these, 1,085 contained at least one Ba31 match. We identified 34 samples with Ba31 counts above the 99% threshold predicted by the BCerG coverage. These samples did not include the two ( P00134 and P00497), previously flagged as *B. anthracis* positive (*Afshinnekoo et al., 2015a*) (Table 3). KrakenHLL also classified each these 34 samples as positive for *B. anthracis*

## DISCUSSION

In this work we have described a significant update to a *B. anthracis* specific 31-mer set that was introduced in earlier blog posts (*Petit III et al., 2015*; *Minot et al., 2015*) and we have shown how this set can be used to interpret *B. anthracis* specific signatures in Illumina metagenome samples. We chose to use k-mer-based signatures for the ease and speed of computation, with the length of 31 nt selected as it was identified as the shortest likely to be unique across bacteria datasets (*Koslicki & Falush, 2016*).

Some species present unusual challenges for metagenome identification. There is no consistently applied definition for the boundary that divides bacterial species based on DNA sequence identity and in some cases the presence or absence of mobile elements like plasmids and phages are required for speciation. *B. anthracis* is closely related to non-biothreat species and acquires its enhanced virulence from genes on mobile plasmids. Such species can be hard to model using "generalist" programs (such as Kraken) that

**Table 2  Artificial mixtures of low coverage *B. anthracis* and high coverage *B. cereus* .**  This table shows some key results from more than 300 artificial mixtures of *B. anthracis* and *B. cereus* sequences created to test our specialized model (File S1). The table includes three *B. anthracis* coverages for each *B. cereus* coverage. The *B. anthracis* simulated coverages represent the minimum *B. anthracis* coverage, the coverage at which *B. anthracis* was detectable, and the maximum *B. anthracis* coverage. The first two columns are the coverage in the artificial mixtures of *B. cereus* and *B. anthracis* genomes, respectively. The third column is the observed BCerG31 k-mer coverage. Columns 4-6 are the observed number of Ba31 k-mers, the expected number of Ba31 k-mers based on the BCerG31 coverage (see Fig. 4) and the 99% prediction interval of the model, which we take as an indicative worst case threshold. The seventh column summarizes whether the observed Ba31 is greater than the 99% P.I. The eighth column is whether the Ba31 coverage is in the "gray zone" ($<0.18\times$ coverage). "No" means the Ba31 exceeds the threshold (note it is possible for the Ba31 coverage to be at gray zone level but still have a positive match to a lef31k-mer). The final column shows whether KrakenHLL (*Breitwieser & Salzberg, 2018*) run on the sample predicted the presence of *B. anthracis*. This table shows that false positives k-mers resulting from high BCerG coverage limit the detection of *B. anthracis* k-mers (Ba31) in mixed cultures. Below $0.006\times$ ($75\times$-fold *B. cereus*) and $0.01\times$ ($100\times$-fold B. cereus) *B. anthracis* genome coverages, true positive Ba31 matches cannot be differentiated from false positive matches. KrakenHLL predicted *B. anthracis* to be present even when it was not because of the background BCerG genomes coverage.

| Artificial genome coverage | | BCerG31 coverage | Ba31 count | | | Exceeds 99% P.I.[a] | lef31 gray zone | KrakenHLL |
|---|---|---|---|---|---|---|---|---|
| *B. cereus* | *B. anthracis* | | Observed | Model fit | Model upper 99% P.I.[a] | | | |
| $0\times$ | $0.001\times$ | 0.00002 | 10 | 1 | 331 | No | Yes | Yes |
| $0\times$ | $0.003\times$ | 0.00245 | 346 | 1 | 332 | Yes | Yes | Yes |
| $0\times$ | $0.2\times$ | 0.123 | 25,396 | 21 | 352 | Yes | No | Yes |
| $1\times$ | $0\times$ | 0.593 | 99 | 102 | 433 | No | Yes | Yes |
| $1\times$ | $0.003\times$ | 0.610 | 444 | 104 | 437 | Yes | Yes | Yes |
| $1\times$ | $0.2\times$ | 0.727 | 25,627 | 125 | 456 | Yes | No | Yes |
| $5\times$ | $0\times$ | 3.048 | 487 | 524 | 855 | No | Yes | Yes |
| $5\times$ | $0.003\times$ | 3.060 | 919 | 526 | 857 | Yes | Yes | Yes |
| $5\times$ | $0.2\times$ | 3.155 | 25,502 | 542 | 874 | Yes | No | Yes |
| $10\times$ | $0\times$ | 6.115 | 1,050 | 1,051 | 1,382 | No | Yes | Yes |
| $10\times$ | $0.004\times$ | 6.100 | 1,531 | 1,048 | 1,379 | Yes | Yes | Yes |
| $10\times$ | $0.2\times$ | 6.450 | 26,346 | 1,074 | 1,405 | Yes | No | Yes |
| $25\times$ | $0\times$ | 15.277 | 2,516 | 2,625 | 2,957 | No | Yes | Yes |
| $25\times$ | $0.004\times$ | 15.174 | 3,075 | 2,608 | 2,939 | Yes | Yes | Yes |
| $25\times$ | $0.2\times$ | 15.339 | 27,536 | 2,636 | 2,967 | Yes | No | Yes |
| $50\times$ | $0\times$ | 30.381 | 5,058 | 5,221 | 5,552 | No | Yes | Yes |
| $50\times$ | $0.005\times$ | 30.438 | 5,726 | 5,231 | 5,562 | Yes | Yes | Yes |
| $50\times$ | $0.2\times$ | 30.595 | 29,766 | 5,257 | 5,589 | Yes | No | Yes |
| $75\times$ | $0\times$ | 45.753 | 7,323 | 4,530 | 8,194 | No | Yes | Yes |
| $75\times$ | $0.006\times$ | 45.699 | 8,351 | 7,853 | 8,184 | Yes | Yes | Yes |
| $75\times$ | $0.2\times$ | 45.859 | 31,971 | 7,881 | 8,212 | Yes | No | Yes |
| $100\times$ | $0\times$ | 60.926 | 9,633 | 10,470 | 10,801 | No | Yes | Yes |
| $100\times$ | $0.01\times$ | 60.958 | 11,020 | 10,475 | 10,807 | Yes | Yes | Yes |
| $100\times$ | $0.2\times$ | 61.093 | 33,761 | 10,498 | 10,830 | Yes | No | Yes |

**Notes.**
[a]Prediction interval.

attempt to classify every read in the dataset into one of thousands of taxonomic groups. We use a "specialist" approach aiming to solve a narrow problem that can be used to augment the predictions of generalist software. Specialist analyses can take advantage of unique features of the system and can also afford more effort in the curation of training

**Table 3  Reanalysis of NYC subway metagenome sequencing.** We counted Ba31, BCerG31 and lef31 k-mers in 1,458 NYC subway metagenomic samples (*Afshinnekoo et al., 2015a*; *Afshinnekoo et al., 2015b*). The table is a breakdown of samples that were within the gray zone and/or had Ba31 matches that exceed the 99% prediction interval. Columns 2–8 display the same data types as columns 3–9 in Table 2. The additional lef column shows whether lef31 matches were identified or not. The final column provides the outcome case of the sample (Table 1). This table presents four samples excerpted from the complete results for all samples (File S2). There is one sample within the gray zone (P00738), two from the original study (P00134 and P00497) and an outlier of samples which exceed the 99% prediction interval (P00981).

| Sample | BCerG31 coverage | Ba31 Count | | | Exceeds 99% P.I.[b] | Gray zone | KrakenHLL | lef | Outcome case |
| | | Observed | Model fit | Model upper 99% P.I.[b] | | | | | |
| --- | --- | --- | --- | --- | --- | --- | --- | --- | --- |
| P00134[a] | 19.71 | 2,755 | 3,387 | 3,718 | No | No | Yes | No | 4 |
| P00497[a] | 4.05 | 953 | 696 | 1,027 | No | No | Yes | No | 4 |
| P00981 | 1.32 | 20,079 | 226 | 558 | Yes | No | Yes | No | 3 |
| P00738 | 0.002 | 396 | 1 | 331 | Yes | Yes | Yes | No | 2 |

**Notes.**
[a]Samples previously identified as containing *B. anthracis*.
[b]Prediction Interval.

data. In this case, we designed 31-mer signature sets based on comparison of hundreds of complete *Bacillus* genomes and we incorporated knowledge of false positive k-mers likely to be produced by close relatives of *B. anthracis* to develop a 'worst case' linear regression model to differentiate *B. anthracis* from sequencing errors. We also used the fact that the presence of a specific gene (*lef*) was diagnostic for anthrax. In designing our k-mer sets we encountered some rare cases of taxonomic mis-assignment in public datasets and were able to take corrective action (Fig. 3). Generalist programs also rely on the same taxonomy and reference sequence databases, but it is harder to detect small errors that lead to mis-assignments when done on a large scale (*Nasko et al., 2018*). If we were to attempt approaches to specifically detect other known *B. cereus* strains that contain pXO1(*Hoffmaster et al., 2004*; *Klee et al., 2010*), we would have to develop and test new k-mer sets based on their unique chromosomal SNPs. Although we concentrate here on *B. anthracis* and BCerG, specialist methods could also be developed for other bacterial pathogens (e.g., *Yersinia pestis* and *Shigella sonnei)* using a similar strategy of accounting for possible non-pathogen close relatives in the sample and the diagnostic presence of high consequence virulence genes acquired by horizontal transfer.

Even when a specialized algorithm has been developed, judgement is still required in interpreting results. In the case of the *Bacillus* genomes in particular, DNA extraction biases may affect results in ways we cannot assess without empirical experiments. We can't tell what proportion of the DNA came from lysis-resistant spores and what proportion was from the more fragile vegetative state, and how this balance might vary between strains across environments. Similarly, using a different sequencing technology, such the Pacific Biosystems SMRT system, with a different error profile, would require recalibration of the model.

Our reanalysis of the NYC data (*Afshinnekoo et al., 2015a*) showed that there was no direct evidence for the lethal factor k-mers in the metagenome samples (File S2). This confirms other work (*Mason, 2015*; *Minot et al., 2015*; *McIntyre et al., 2017*), and together

with the low prior probability of encountering *B. anthracis* in New York City, suggests that the samples taken were all negative for anthrax. The two samples originally flagged as possibly positive (Table 3) fell under case 4 (Table 1), as did 1,049 out of the other 1,456 samples. There were 373 samples with no Ba31 k-mer matches. These are all most likely true negatives, although, as we showed in the synthetic dataset, high BCerG coverage can mask the signal of low coverage *B. anthracis* (Table 2). To get a true negative would theoretically involve sequencing every cell in the sample (assuming perfectly efficient DNA preparation), which is impossible currently for all but the simplest communities. The limit of detection will be a complex calculation that involves the amount of DNA sequence generated and the complexity of the microbial community. Negative (and positive) calls ultimately have to be supported through sensitive detection assays such as PCR and/or culture.

We identified 34 samples above the BCerG thresholds for our model (Table 3). All the samples fell under case 3 except a single sample which fell under case 2 (Table 1). An outlier of case 3 samples, P00981, taken from a metal handrail on the A train route (*Afshinnekoo et al., 2015a*), had high Ba31 counts ($n = 20,079$). As we collect more genomes of *B. cereus* group we may see more Ba31 k-mers in BCerG genomes. These samples may contain members of yet unencountered lineages more closely related to *B. anthracis* than previously seen, or possibly the result of recent recombination between *B. anthracis* and *B. cereus* genomes (although the latter has not been reported). It is important that these strains are isolated, sequenced and added to public databases to iteratively improve pathogen detection. The single case 2 sample, P00738 (Table 3), was also on a metal handrail from the A train route, although sampled 3 days earlier than P00981. This sample was possibly the most problematic because the Ba31 counts were in the gray zone, meaning there was not enough coverage to rule out *lef* being present. Most likely, this sample contained another near-*B. anthracis* strain, but case 2 samples should be a priority for retesting by culture and PCR methods.

## CONCLUSIONS

If *B. anthracis,* or another BCerG strain containing pXO1, is present in a shotgun metagenome sample at high genome coverage, identification of *lef* k-mers is a strong signal for the likely presence of anthrax-causing bacteria. We showed that using a *B. anthracis* specific k-mer set alone to call the presence of *B. anthracis* produced many false positive calls because sequencing errors of common co-resident BCerG bacteria. We developed models to partition cases that contained evidence of possible low coverage *B. anthracis*, accounting for *B. cereus* coverage. However, in simulations, we showed that false negative results can arise when the BCerG coverage is high. Reanalysis of the NYC subway metagenome study confirmed the absence of *B. anthracis* containing *lef* but we found evidence in at least two samples of BCerG strains that contained what were considered *B. anthracis* specific sequences. Culturing strains such as these, genome sequencing and sharing to the public domain will help improve *B. anthracis* detection in metagenome shotgun samples.

## ACKNOWLEDGEMENTS

Thanks to Sam Minot, Chris Greenfield, Chris Mason and his group for discussion following our original blog post.

### Funding

This study has been supported by development funds to Timothy D. Read from the Emory University School of Medicine and the Seven Bridges NCI Cancer Genomics Cloud Pilot, supported by funds from the National Cancer Institute, National Institutes of Health, Department of Health and Human Services, under Contract No. HHSN261201400008C. There was no additional external funding received for this study. The funders had no role in study design, data collection and analysis, decision to publish, or preparation of the manuscript.

### Grant Disclosures

The following grant information was disclosed by the authors:
Emory University School of Medicine and the Seven Bridges NCI Cancer Genomics Cloud Pilot.
National Cancer Institute, National Institutes of Health.
Department of Health and Human Services: HHSN261201400008C.

### Competing Interests

Timothy D. Read is an Academic Editor for PeerJ.

### Author Contributions

- Robert A. Petit III conceived and designed the experiments, performed the experiments, analyzed the data, contributed reagents/materials/analysis tools, prepared figures and/or tables, authored or reviewed drafts of the paper, approved the final draft.
- James M. Hogan and Timothy D. Read conceived and designed the experiments, analyzed the data, prepared figures and/or tables, authored or reviewed drafts of the paper, approved the final draft.
- Matthew N. Ezewudo and Sandeep J. Joseph performed the experiments, contributed reagents/materials/analysis tools, approved the final draft.

### Data Availability

Robert A. Petit III. rpetit3/anthrax-metagenome-study: Repo state as of July 30th, 2018 (Version final-revisions). Zenodo. http://doi.org/10.5281/zenodo.1323741.
Code repository of previous analysis: https://github.com/Read-Lab-Confederation/nyc-subway-anthrax-study.

## Supplemental Information

Supplemental information for this article can be found online at http://dx.doi.org/10.7717/
peerj.5515#supplemental-information.

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
