# Peer review of "Fine-scale differentiation between Bacillus anthracis and Bacillus cereus group signatures in metagenome shotgun data"

_PeerJ, doi:10.7717/peerj.5515_

## Round 0.1 · original submission · Minor Revisions

Please modify the manuscript in line with the reviewers' comments. I agree with their combined efforts and have no specific comments of my own to add. Echoing the sentiments of two of the referees and to promote reproducible research, I request all scripts/tools used in this study are made publicly available before the revised manuscript is submitted to PeerJ, and that basic parameters used for programs are included in Methods.

·

Basic reporting

The manuscript is clearly written and well structured, and meets the journal's criteria.

Experimental design

1. There are now many classifiers available which vary in their specificity and sensitivity, recently reviewed by McIntyre et al (as cited in the manuscript). McIntyre also specifically addresses this question of identification of B. anthracis in these same samples, and reports that when combining tools to reduce false positives, 'two of six such pairs still reported anthrax', with the focus on pointing out that false positives are a problem, but does not elaborate on the 4 of the six pairs which do not report anthrax. While the authors of this manuscript include comparisons with Kraken and MetaPhlAn results (from the Afshinnekoo paper), a direct comparison with the findings of McIntyre with relation to these samples would seem appropriate to gaining a fuller understanding of the limitations and utility of the different methods.

2. A comparison of the results of this analysis with KrakenHLL was performed using 'the standard Kraken database'. A number of different pre-built databases are currently available for Kraken (4Gb and 8Gb minikraken databases which available with and without DUST masking: https://ccb.jhu.edu/software/kraken/). The use of a DUST-masked database should significantly reduce the false positive rate, however it is not clear from the manuscript which of these databases was used. This should be clarified, and results from using a DUST masked database should be provided. Since the authors describe the identification of a number of misclassified B. cereus genomes in the databases, a better comparison with Kraken would be to use a custom database including these taxonomical corrections.

3. The authors have gone to admirable lengths in providing access to scripts and code used to carry out the analysis described in this paper, however identifying precisely how a given program was run requires the reader to locate the appropriate script in the github repository then find the appropriate command-line in the script. It would be preferable if basic parameters of programs (i.e. which bwa algorithm?) used in the analysis were included directly in the methods

4. A number of Entrez queries used to download reference data are provided, however the results of these queries are highly variable with time. Providing a list of the accessions (including sequence version) of the sequences included in these reference sets in the github repository would greatly assist anyone attempting to replicate or build on this analysis.

Validity of the findings

No comment

Additional comments

This is a paper which clearly demonstrates the dangers associated with making assumptions on the basis of general metagenomics classification methods. It is weakened by only comparing results with Kraken and MetaPhlan, while the data of McIntyre et. al is suggestive of reduced false positive rates when using certain classifiers in combination.

The paper would also benefit from discussion of how readily this approach can be adapted for other organisms.

·

Basic reporting

A clear, unambiguous, professional English language was mostly used throughout.
I highlighted specific cases where I could imagine that appropriate rephrasing would benefit the readability of the paper.
The intro & background show the necessary context and refer to the relevant literature.
I have not checked whether the word counts are followed but leave this up to the journal to verify. The structure appears to conform to PeerJ standards.
The figures are relevant, high quality, and well labeled.
While for the main figures the respective captions were given, I could not find the captions of the Supplementary Figures.
Raw data has been supplied based on a high-level check. I have not been able to check all details.
The availability of an Rmarkdown based notebook on Github is commendable.

Experimental design

The original primary research presented in this work is within the scope of the journal.
The research question is well defined, relevant & meaningful. It is stated how the research fills an identified knowledge gap.
Rigorous investigation was performed to a high technical & ethical standard.
The methods are described largely with sufficient detail, please also see my general comments for cases which could be improved. While I have not attempted to fully replicate the findings based on the information available, the authors appear to have followed very good practices by using version control (via Github) and provide an Rmarkdown file including code.

Validity of the findings

The data appears robust, statistically sound, & controlled
The conclusions are well stated, linked to the original research question & limited to supporting results.

Additional comments

SUMMARY
Petit et al. present an interesting study on the identification of biothreat-derived sequences from metagenomic data, focused on the biothreat agent Bacillus anthracis.
The authors investigated possible sources for errors in the taxonomic assignment of metagenomic samples. In order to overcome these errors, the authors propose a gene-based (i) and a genome-wide (ii) approach to predict the presence of pathogenic species, as well as a sequencing-coverage dependent model (iii) to reduce the likelihood of false-positive B. anthracis assignments. Moreover, the authors reanalysed data from a study which was strongly and openly discussed in the scientific community and for which respective corrections were originally performed. The present work further supports this and it is commendable that the authors did it in this structured and extensive way.
The manuscript is overall well written. However, I was lost sometimes (indicated in my general comments below) and I think that the authors should try to improve these parts of their manuscript in order to improve its readability.

GENERAL COMMENTS:
1. L20: "small phylogenetic distance and both the pathogen and non-pathogen species are
present in the same sample". I am not sure about the use of "phylogenetic" distance in this context. While I understand what the authors mean, a species may become pathogenic with the acquisition of only a few genes, yet, the phylogenetic distance would be extremely small, maybe even zero. Moreover, it depends on which _gene_ is used to define the "distance": In case of one of the acquired pathogenicity-related genes, the distance would be large. This could be rephrased to reflect a rather evolutionary perspective, maybe.
2. L27: "we identified 31-mer signatures that differentiated B. anthracis from other members of the B. cereus group (BCerG), and from 31-mers conserved in all BCerG genomes (including B. anthracis), but not in other Bacillus strains". This sentence should be rephrased for clarity, e.g., "[...] and we identified 31-mer signatures that differentiated BCerG genomes from other Bacillus strains", given that I correctly understood the original intention of the authors here.
3. L73: "ANI (average nucleotide identity)". Isn't it a convention to first write the text out before introducing the abbreviation? The authors did so at least for the k-mer sets in the abstract. Maybe this could be adjusted accordingly.
4. L88: "Plasmids from other BCerG genomes may be very similar to pXO1 and pXO2, but lack the important virulence genes." Is this a speculation or do the authors have a reference for this? If so, please include it as concrete examples of this would further support the importance of the authors' work.
5. L113: Could the authors please explain the use of "NOT txid86661[Organism:exp]", i.e., why were B. cereus genomes excluded here?
6. L141: "from from"
6.1. L152: While this is clearly a debatable point, I was wondering why did the authors include Suppl. Fig. 1 into the Supplements instead of the main text? I think this information can help the reader to more readily understand the individual steps. Moreover, I think that the k-mer set names, e.g., Ba31, could be included as output of the workflow to link this better to the manuscript text.
7. L184: Were the simulated reads error-trimmed or pre-processed somehow? I am not sure about how ART simulates the reads, but Illumina sequencers are know to have an increasing drop in quality (and hence increase in the error) towards the end of the reads. The number of erroneous k-mers would thus also be expected to increase towards the reads' ends.
8. L185: I understand that a certain model had to be fixed here. Could the authors please elaborate on why the HiSeq2000 error model was chosen? Was it the "most recent" one available in ART? Is it assumed that the HiSeq2000 is the most common sequencing instrument? How about newer Illumina machines? Depending on the amount of extra work for this, it might be good to compare the performance of the k-mer sets on the HiSeq2000 error model with the performance on NextSeq500, for example representing a newer Illumina machine.
9. L189: How about "precision"? Moreover, the authors refer to these performance metrics but seem to not use them later in the manuscript, at least not in their mathematical sense. I would suggest revising this accordingly.
10. L194: Please also report the R version and version of the base stats package.
11. L198: Does the title refer to "host-derived/environmental samples" as the text mentions "metagenomic mixtures"? If so, please include this in the title of L198.
12: L204: What is the reason for not using the k-mer set name "Ba31" here? I would find it more consistent to stick to these namings as much as possible throughout the manuscript.
13: L210: Please see my comment about L204.
14: L252: I could not readily see why "~0.18x" would be the cut-off. The highlighted fold-coverage in Fig. 1 do not match this number. As it seems to be an approximation ("~"), it would be good to specify the exact value at least once.
15: L257: I am unsure if I have missed something here. Because, while I understand what the authors mean here, from my reading of the "previous section", it was about the lef gene, not about B. anthracis chromosomal markers in general. I would hence suggest to rephrase this sentence accordingly or include results on chromosomal markers (beyond the plasmid-borne lef gene).
16: L271: I found this to be very interesting. It highlights the importance of taxonomic curation; in general but also within the context of this specific approach.
17: L276: It was not readily clear to me where the "coverage" comes from, i.e., which data is searched for the respective Ba31 and/or BCerG k-mers. Is it still the NYC subway metagenomic dataset? Or is it the 164 B. anthracis genome projects? Or is it the set described in L191 ff? This is a crucial point as the (linear) model performance is expected to be highly dependent on the training/learning of the model parameters. Please make this more clear.
18: L286: Is this shown somewhere, i.e, the results of non-BCerG genomes?
19: L292: While I understand the rationale for subsetting the BCerG strains to a clade that is different, yet most closely related to B. anthracis, I think that the emerging linear model is then somewhat misleading. Concretely, it is not clear whether the model will be applicable to BCerG in general, or rather to this specific clade. In case of the former, an initial "closeness check" would be needed to test if the conditions/assumptions of the developed model are fulfilled or not. Interestingly, the authors rightly state this in L308. I would suggest to emphasize this more strongly, potentially even in the Abstract.
20: L302: An "in" is missing -> "is shown _in_ Table 3 ..."
21: L349: I would suggest to also include the developed sequencing error model into the list of "outputs". Or is there a particular reason that the authors have omitted it?
22: L390: "is" is missing -> "It _is_ important ..."
23: L401: This reads a bit contradictory to me. If the k-mer set was _specific_ to B- anthracis, why would one expect the false positive calls to be due to common co-resident BCerG bacteria, rather than due to sequencing errors, especially in case of high-coverage B. cereus members? Please let me know, should I have missed something there.

·

Basic reporting

Overall this is a good summary of the challenges and new methods that have emerged for delineating various strains of Bacillus, and provides a new framework for separating the closely related strains. This manuscript is well-structured and provides useful tools for future work.

A few questions about the current manuscript include:

1) They used the Mash distance to reclassify each Bacillus genome as B. anthracis (Mash distance <= 0.000687), non-anthracis BCerG (Mash distance <= 0.135333), or non-BCerG (Mash distance > 0.135333). But is the Mash threshold the best measure? I am wondering if these results would change if the thresholds were targeted to specific regions of the genome, or the plasmids, and if this could improve the separation?
2) They have built models on 100x100 reads, but can the newer 150x150 reads improve the resolution? I know this would be a bit of work but it would give more staying power to the analysis since almost all reads today at 150x150, and it may help with resolution of the Ba strains.
Small issues:
1) There is a missing period at the end of the first paragraph.
2) Is their python script (subsample-ba-lod.py) and other tools used publicly available? I don’t see it in the Zenodo link (for which the authors are commended, BTW), since some of the links (ilke bam2fastq) are no longer viable, and/or have been deprecated for new tools (like Picard) from https://gsl.hudsonalpha.org/information/software/bam2fastq.
3) The authors should also and the official correction from the journal: https://www.cell.com/cell-systems/fulltext/S2405-4712(15)00015-0.

Experimental design

Find as is.

Validity of the findings

An update to a hard question in species classification and useful sets of k-mer tools are set up as well.

---

## Round 0.2 · accepted · Accept

I am happy that you have addressed the reviewers' specific comments, and look forward to seeing your published article in the near future.

#